# Antimicrobial Exposure in Critically Ill Patients with Sepsis-Associated Multi-Organ Dysfunction Requiring Extracorporeal Organ Support: A Narrative Review

**DOI:** 10.3390/microorganisms11020473

**Published:** 2023-02-13

**Authors:** Salvatore Lucio Cutuli, Laura Cascarano, Paolo Lazzaro, Eloisa Sofia Tanzarella, Gabriele Pintaudi, Domenico Luca Grieco, Gennaro De Pascale, Massimo Antonelli

**Affiliations:** 1Dipartimento di Scienze dell’Emergenza, Anestesiologiche e Della Rianimazione, Fondazione Policlinico Universitario A. Gemelli IRCCS, L.go A. Gemelli 8, 00168 Rome, Italy; 2Dipartimento di Scienze Biotecnologiche di Base Cliniche Intensivologiche e Perioperatorie, Universita’ Cattolica del Sacro Cuore, Rome, L.go F. Vito 1, 00168 Rome, Italy

**Keywords:** sepsis, multi-organ dysfunction, extracorporeal organ support, extracorporeal blood purification therapy, renal replacement therapy

## Abstract

Sepsis is a leading cause of disability and mortality worldwide. The pathophysiology of sepsis relies on the maladaptive host response to pathogens that fosters unbalanced organ crosstalk and induces multi-organ dysfunction, whose severity was directly associated with mortality. In septic patients, etiologic interventions aiming to reduce the pathogen load via appropriate antimicrobial therapy and the effective control of the source infection were demonstrated to improve clinical outcomes. Nonetheless, extracorporeal organ support represents a complementary intervention that may play a role in mitigating life-threatening complications caused by sepsis-associated multi-organ dysfunction. In this setting, an increasing amount of research raised concerns about the risk of suboptimal antimicrobial exposure in critically ill patients with sepsis, which may be worsened by the concomitant delivery of extracorporeal organ support. Accordingly, several strategies have been implemented to overcome this issue. In this narrative review, we discussed the pharmacokinetic features of antimicrobials and mechanisms that may favor drug removal during renal replacement therapy, coupled plasma filtration and absorption, therapeutic plasma exchange, hemoperfusion, extracorporeal CO_2_ removal and extracorporeal membrane oxygenation. We also provided an overview of evidence-based strategies that may help the physician to safely prescribe effective antimicrobial doses in critically ill patients with sepsis-associated multi-organ dysfunction who receive extracorporeal organ support.

## 1. Introduction

Sepsis is a heterogenous clinical syndrome characterized by host and pathogen factors [1] and represents a major cause of disability and mortality worldwide [2]. Sepsis originates from a maladaptive response to infection leading to unbalanced organ crosstalk and multi-organ dysfunction (MOD), whose severity was directly associated with clinical complications and mortality [1]. The burden of sepsis may be mitigated by etiologic interventions that aim to reduce the pathogen load via therapeutic exposure [3] to appropriate antimicrobials (that is the antimicrobial-free fraction concentration matching pathogen susceptibility) [4] and effective control of source infection [5]. However, “standard” dosing regimens based on empiric rules in critically ill septic patients were associated with inadequate antimicrobial exposure, which was mostly due to sepsis-associated homeostatic alterations affecting the drug distribution and clearance [3,6,7], and implied an increased risk of worse clinical outcomes [3].

Moreover, inadequate antimicrobial exposure may be fostered by the concomitant delivery of extracorporeal organ support (ECOS) [8,9], which includes several blood purification therapies widely used in clinical practice to mitigate the life-threatening complications of MOD (e.g., hyperuremia, fluid overload and acid base imbalances) [10,11]. In this setting, several studies raised concern about the suboptimal antimicrobial exposure associated with ECOS [12] leading to enhanced risk of treatment failure and the emergence of multi-drug resistant (MDR) pathogens [13,14]. For these reasons, the 2021 Surviving Sepsis Campaign (SSC) guidelines recommended administering empiric antimicrobials immediately after sepsis recognition and optimizing dosing strategies based on pharmacokinetic/pharmacodynamic (PK/PD) principles, with the aim of maximizing pathogen killing and preventing drug toxicity [15]. In the light of this view, several antimicrobial dosing strategies have been proposed and include unit-level interventions (e.g., prolonged infusion), nomograms or therapeutic drug monitoring (TDM) [6,16,17].

In this narrative review, we highlighted the PK alterations of antimicrobials induced by sepsis and ECOS. Moreover, we focused on specific antimicrobial properties that may favor drug removal via ECOS. Finally, we provided an overview of strategies that may help the clinician to overcome the issue of inadequate antimicrobial exposure during ECOS in critically ill patients with sepsis-associated MOD.

## 2. PK/PD Considerations

Antimicrobial therapy is one of the most common pharmacologic interventions prescribed in the ICU [18,19]. In order to maximize pathogen-killing and avert drug toxicity, antimicrobial dosing strategies should be oriented by a personalized approach based on PK/PD principles [16,20] and not standardized on empiric rules neglecting patient demographic and clinical characteristics, pathogen features and antimicrobial properties (Figure 1).

The PK profile of an antimicrobial relies on the interaction between patient characteristics (e.g., body composition or organ function), concurrent interventions (e.g., ECOS) and drug properties (physiochemical features and clearance, Table 1) affecting the time-course concentration of the antimicrobial in specific tissues [21]. In contrast, the PD profile refers to the antimicrobial mechanism of action (Figure 1). According to PK/PD principles, an effective antimicrobial dose should deliver an adequate antimicrobial-free fraction concentration at the source of infection, in order to match pathogen susceptibility expressed using the minimum inhibitory concentration (MIC) [22]. The MIC is determined in vitro and represents the antimicrobial concentration (twofold dilutions up and down starting from 1 mg/L) that prevents visible growth over 18 + 2 h at 34–37 °C, using a standardized inoculum, thus representing the bacteriostatic/bactericidal effect of the drug over time [23]. However, the MIC measurement is burdened by biological (strain-to-strain differences) and laboratory (e.g., assay used, inoculum preparation, incubation temperature and time, technical skills) variations [23]. In order to overcome the heterogeneous assessment of MIC among different laboratories, the breakpoints for phenotypic pathogen susceptibility to antimicrobials are determined by regulatory agencies such as the European Committee on Antimicrobial Susceptibility Testing (EUCAST) [24] that revise and publish updated tables yearly based on a complex scientific process that accounts for: the target infection, drug (e.g., dosage, mode of administration, PK and PD) and pathogen (e.g., resistance mechanisms) characteristics, disk diffusion methods and zone diameter distributions, epidemiological cut-off values, MIC distributions and clinical outcomes. The EUCAST antimicrobial susceptibility testing breakpoints are reported as MIC breakpoint values (mg/dl) and zone diameter breakpoint values (mm), and graded as:−S—susceptible, standard dosing regimen: high likelihood of therapeutic success using a standard dosing regimen of the antimicrobial;−I—susceptible, increased exposure: high likelihood of therapeutic success if antimicrobial exposure is improved by increasing the dosing regimen in order to reach high drug concentration at the site of infection;−R—resistant: high likelihood of therapeutic failure even for increased exposure.

Consequently, higher antimicrobial exposure is required for the treatment of less susceptible strains and, whenever the measured MIC of a pathogen causing a specific infection (e.g., pneumonia, bacteremia, meningitis, etc) is above the MIC breakpoint indicating antimicrobial resistance, some author suggested to target an antimicrobial exposure at the source of infection equal to 4 x pathogen MIC, unless the drug dose exceeds the threshold of toxicity [25]. 

### 2.1. PK Alterations Induced by Sepsis

Antimicrobials are characterized by specific properties, whose interaction with sepsis-induced MOD influence their distribution and time-course concentration in different tissues:➢Solubility is the major determinant of antimicrobial volume of distribution (Vd), which is the theoretical volume necessary to contain the total amount of the drug at the same concentration measured at the steady state in the plasma. The Vd drives the titration of the loading dose of antimicrobials [26]. Specifically, hydrophilic antimicrobials (e.g., ß-lactams, aminoglicosides, glycopeptides, daptomycin and polymyxins) are characterized by small Vd (<2 L/kg), being mainly concentrated into the bloodstream. However, sepsis-associated endothelial dysfunction and capillary leak syndrome coupled with fluid overload due to large fluid resuscitation and oliguric acute kidney injury (AKI), may lead to extravascular fluid shift and, consequently, low bloodstream concentration of hydrophilic antimicrobials. This condition implies an increase in the loading dose in order to secure an effective exposure to these drugs. In contrast, lipophilic antimicrobials (e.g., fluoroquinolones, glycocyclines, lincosamides, macrolides) are characterized by large Vd (>2 L/kg) and are not significantly influenced by fluid shift, being mainly concentrated in the adipose tissue. ➢Protein binding varies among antimicrobials (Table 1). The main carriers of antimicrobials into the bloodstream are represented by albumin for acidic drugs (e.g., aminoglycosides and levofloxacin) and α1-acid glycoprotein for basic drugs [21], in accordance with to the Gibbs–Donnan effect. ➢In critically ill patients, hypoalbuminemia has been frequently reported [27], and this condition may alter the PK profile of highly-protein bound antimicrobials (e.g., ß-lactams, glycopeptides, glycylcyclines, lincosamides, daptomycin and macrolides), leading to augmented free fraction, Vd and clearance.➢Electrostatic interactions regulate the degree of antimicrobial ionization and free fraction, the amount of which relies on the pKa of the drug. Specifically, the free fraction of weak basis-like antimicrobials is increased by pH of tissues below pKa, while the dissociation of weak acid-like antimicrobials is favored by local pH above the pKa [21]. ➢Molecular size influences drug excretion. Small molecules are preferentially filtered by the kidney into urine and large molecules are secreted by the liver into the bile [21]. ➢Clearance refers to the volume of plasma purified from antimicrobials per unit of time and drives the titration of the maintenance dose [26]. Catabolism and excretion of hydrophilic antimicrobials take place predominantly in the kidney (e.g., ß-lactams, aminoglucosides, glycopeptides, daptomycin, oxazolidones, polymyxins), while the liver plays a role of paramount importance for lipophilic antimicrobials (e.g., glycylcyclines and macrolides). Sepsis-associated MOD may induce alterations of antimicrobial metabolism [21] leading to suboptimal exposure or intoxication. In the early phases of sepsis, hyperdynamic states due to cardiac output increase may enhance glomerular filtration rate, leading to augmented renal clearance [6,14] and suboptimal antimicrobial exposure when prescribed at standard dosing regimens. On the contrary, sepsis is the major cause of AKI [28], and this condition may increase the risk of antimicrobial accumulation and toxicity. 

### 2.2. PK Alterations Induced by ECOS

ECOS represents an additional PK compartment that may influence antimicrobial exposure by altering the distribution and clearance of these drugs. The delivery of ECOS implies an unavoidable administration of intravascular volume due to priming solution, which increases Vd and lowers the bloodstream concentration of hydrophilic drugs [20]. Moreover, antimicrobials may be cleared via ECOS (Table 2) through absorption or trans-membrane removal (Table 2). Absorption is influenced by the biochemical interaction of the drug with the internal surface of the device, and represents the main mechanism of antimicrobial removal during hemoperfusion, coupled plasma filtration and absorption (CPFA), extracorporeal CO_2_ removal (ECCO2-R) and extracorporeal membrane oxygenation (ECMO). Trans-membrane removal is regulated by concentration gradients (haemodialysis, HD), pressure gradients (hemofiltration, HF) or both (hemodiafiltration, HDF) along the device [10,11]. Trans-membrane removal relies on the interaction between the physiochemical properties of the drug (e.g., solubility and protein binding) and the membrane (e.g., material and permeability), the technique (e.g., HD, HF, HDF) and its setting (e.g., dose). This is the leading mechanism of antimicrobial removal during renal replacement therapy (RRT), CPFA and therapeutic plasma exchange (TPE).

## 3. Potential Pitfalls in the Evaluation of ECOS-Related Antimicrobial PK Alterations 

The majority of research on PK alteration of antimicrobials induced via ECOS is represented by small case series not accounting for the endogenous clearance (residual kidney and liver function) [36], thus implying low external validation and reproducibility. In this context, a potentially useful method to assess residual renal function in severe critically ill patients may be represented by the measurement of the 2-h creatinine clearance [37]. 

Moreover, most of the studies in this field are represented by pre-clinical investigations using mechanistic ex vivo one-compartment models, whose translation into the clinical scenario may be limited by several factors. First, ex vivo models do not account for the endogenous clearance of antimicrobials. Second, antimicrobials are dissolved in surrogate solutions of patient’s blood (e.g., crystalloids, albumin or plasma) that may alter the PK characteristics of the drug (e.g., solubility and free fraction). Third, ex vivo models evaluate the effect of ECOS on a single antimicrobial dose, thus underestimating drug accumulation and device saturation for repeated antimicrobial administration, both mitigating the impact of ECOS on antimicrobial exposure in vivo. For these reasons, in our review, we preferentially reported the results of clinical studies over ex vivo models and mentioned the latter if we could not retrieve evidence in vivo.

## 4. Renal Replacement Therapy

Sepsis is the major cause of AKI in the ICU [28]. Specifically, the 15% of critically ill patients with septic AKI require RRT [38], and these patients are burdened by 50% higher mortality than patients on RRT without septic AKI [39]. The SMARRT trial [40] was a large multicenter study (29 ICUs from 14 countries) that investigated antimicrobials’ PK (piperacillin, tazobactam, meropenem and vancomycin) during CRRT, and reported a considerable variation of RRT modes and intensities, total estimated renal clearance (defined as total RRT effluent rate and patient’s intrinsic glomerular filtration rate) and antibiotic dosing among the 381 critically ill patients included. Moreover, this study revealed that a “standard” dosing regimen failed to achieve the target antimicrobial exposure in ≥25% of patients and was associated with increased mortality. Accordingly, the SMARRT trial [40] demonstrated that ensuring target antimicrobial exposure by the use of empirical dosing regimens is challenging and implied drug dosing adjustment in response to the real time measurement of antimicrobials. In critical care practice, most of the recommendations on antimicrobial dosing are outdated in comparison with the advancing RRT technology, or are extrapolated from reports in non-critically ill patients with stable chronic kidney disease on intermittent RRT [29]. In this setting, the extracorporeal clearance of an antimicrobial is commonly considered significant when it is higher than the 25% of the total body clearance [26]. In this review we focused on CRRT, which is the most common RRT modality used in European and Australian ICUs, although its use has been increasing in the United States [41]. Specifically, we discussed antimicrobial characteristics that may enhance drug removal during CRRT and provided scientific evidence of this phenomenon.

### 4.1. Membrane Characteristics

The material and internal surface area of CRRT membranes may influence the amount of antimicrobial removal via absorption, which is greater for polyacrylonitrile compared with polysulfone or polymethylmetacrylate membranes (e.g., for gentamycin and tigecycline [42]). Moreover, Ulldemolins et al. [43] reported that larger doses of piperacillin-tazobactam were required to achieve adequate bloodstream concentration for acrylonitrile 69 surface-treated (AN69ST internal surface: 1.5 m^2^) membranes compared with the AN69 (internal surface: 0.9 m^2^). However, limited evidence exists on the clinical impact of this phenomenon, especially for newly developed membranes (e.g., the AN69ST [44]). 

Membrane permeability to solutes is expressed by the Sieving coefficient (SC) for hemofiltration and the saturation coefficient (SA) for hemodialysis, which is about 30–40% less than SC. The SC refers to the ratio of solute concentration in the ultrafiltrate to solute concentration in the bloodstream, while the SA corresponds to the ratio of solute concentration in the dialysate to solute concentration in the bloodstream [45]. Accordingly, trans-membrane solute removal is negligible for SC or SA equal to 0 and is maximal for SC or SA equal to 1. These coefficients are dynamic and decrease proportionally to the duration of each treatment session and the filtration fraction (effluent flow rate/plasma flow rate) [46], due to progressive clotting and the stratification of protein cake on the internal surface of the membrane [11]. Moreover, trans-membrane antimicrobial removal with low-flux membranes (ultrafiltration coefficient, K_UF_ < 12 mL/mmHg/h) is negligible for molecular weight >1000 Da, while it is relevant with high-flux membranes (K_UF_ >12 mL/mmHg/h, mean pore radius 3.9 nm, cutoff around 20–30 KDa) and may be even greater for medium (mean pore radius 5 nm) and high-cutoff (mean pore radius 10 nm, cutoff above 60 KDa) membranes.

### 4.2. Setting

Hydrophilic drugs with small molecular weight (<500–1000 Da), low protein binding (<80%) and small Vd (<2 L/Kg) are theoretically susceptible to trans-membrane removal during HD [29,30]. In contrast, drugs with larger molecular weight could be efficiently removed via HF or HDF. The SMARRT trial [40] reported a large variability of tazobactam and vancomycin bloodstream concentrations in relation to the type of CRRT, with a higher rate of removal for CVVHF and CVVHDF than for CVVHD.

The effluent flow rate (intensity) refers to the volume of blood cleared from solute by the extracorporeal circuit per unit of time and body weight (mL/kg/h) [10,11]. The optimal dose of CRRT in critically ill septic patients with AKI has been debated over the last decades and post hoc analyses of large randomized clinical trials revealed significant variability of antimicrobial concentration during CRRT, possibly due to the adoption of different empiric dosing strategies and CRRT intensities among participating centers [47]. For this reason, no specific “standard” antimicrobial dosing strategy may be recommended to match all forms and prescriptions of RRT. In 2014, Jamal et al. [48] conducted a systematic review and meta-analysis to investigate the PK alterations associated with the different RRT modalities and settings of antipseudomonal β-lactams (such as meropenem and piperacillin) and vancomycin. This study included 30 original articles (349 patients) and found a direct correlation between the CRRT dose and extracorporeal elimination of these drugs. In contrast, a post hoc pharmacokinetic analysis of the RENAL trial [49] failed to demonstrate a dose-dependent relationship of antimicrobial removal (ciprofloxacin, meropenem, piperacillin, tazobactam and vancomycin) between 40 mL/kg/h and 25 mL/kg/h). Nonetheless, the IVOIRE study [50] found that a higher CRRT intensity (70 mL/kg/h) significantly shortened antimicrobial half-lives compared with a lower CRRT intensity (35 mL/kg/h), although antimicrobials were not specified and residual renal function was not assessed.

Recently, Backdach et al. [51] systematically reviewed the results of 27 articles on PK/PD characteristics of novel β-lactams and β-lactams/β-lactamase inhibitors in critically ill patients receiving ECOS. Specifically, they found that increasing doses of Cefiderocol (standard dose: 2 g Q8h) might be necessary to attain PK/PD indexes in relationship with residual renal function, highly resistant pathogen and CRRT intensity (from 1.5 g Q12h for effluent flow rate ≤ 2 L/h to 2 g Q8h for effluent flow rate ≥ 4 L/h), while the maintenance dose of Ceftolozane/Tazobactam should be lowered (from 1.5–3 g Q8h to 0.75 Q8h) during CRRT. In contrast, no evidence was available for other antimicrobials such as Caftazidime/Avibactam, Imipenem/Relabatam and Meropenem/Vaborbactam [51]. 

### 4.3. Central Venous Catheters Tip Location

The Kidney Disease Improving Global Outcome (KDIGO) guidelines [52] recommended to place the central venous dialysis catheter in the superior cava vein district as preferential choices: right jugular vein (first choice), femoral vein (second choice), left jugular vein (third choice) and subclavian vein (last choice). However, some experimental and pre-clinical studies demonstrated the influence of the distance between the central venous catheter (used for drug infusion, central venous pressure monitoring or blood withdrawal) and the central venous dialysis catheter on drug removal. Specifically, they found that drug aspiration was minimal when the central venous catheter tip was placed 0–2 cm distally to the central venous dialysis catheter tip and drug aspiration was maximal when withdrawing the former proximately to the latter at a distance of 4 cm [53,54]. Moreover, the drug infusion speed may affect the amount of drug aspiration much more than the RRT blood flow rate [53]. In contrast, an animal model demonstrated that bloodstream concentrations of drugs (noradrenaline and gentamycin) were less affected via RRT when the central venous catheter and central venous dialysis catheter tips were placed in different central venous districts (e.g., internal jugular vein and femoral vein) [55]. However, no research has assessed the impact of catheter tip location on antimicrobial removal in a clinical setting.

## 5. Therapeutic Plasma Exchange

TPE allows patient’s plasma removal and replacement with human donor plasma, colloids, crystalloids or a mixture thereof, with the aim to improve immune dysfunction and coagulation abnormalities [31]. TPE is mainly used for specific clinical conditions (e.g., myasthenia gravis, Guillain–Barré syndrome and Waldenström macroglobulinemia) [31], although recent studies reported some hemodynamic improvement associated with this therapy in septic patients [56,57]. Low Vd and high protein binding (>80%) may enhance antimicrobial removal during TPE, while molecular size has no role in this setting [31]. Moreover, antimicrobial removal may be increased by the replacement of large plasma volume and drug administration immediately before the TPE session [31]. Recently, Krzych et al. [58] reviewed the current literature and found that ß-lactams (cefazolin and ceftriaxone) and liposomal Amphotericin B are significantly removed via TPE. However, such evidence came from pre-clinical models or small case series, thus warranting future clinical investigation. 

## 6. Coupled Plasma Filtration and Absorption

The CPFA combines plasma filtration and adsorption via styrene resin. Briefly, the volume of plasma purified via mediators is reinfused into the blood line, which passes across an hemofilter [44]. 

CPFA was demonstrated to significantly lower the bloodstream concentration of colistin, whose amount was directly proportional to the volume of plasma filtered over time [32]. Moreover, a retrospective study [59] demonstrated large adsorption of piperacillin, vancomycin and fluoroquinolones on the resin cartridge, which decreased over time and was not associated with altered antimicrobials’ bloodstream concentration. In contrast, minimal absorption was reported for Ceftriaxone, Cefotaxime and Ceftazidime [59]. Evidence of adsorption of other antimicrobials during CPFA is limited and further investigations are warranted in this field. 

## 7. Hemoperfusion

Several devices have been implemented in clinical practice with the aim to remove specific mediators in sepsis [12,44,60]. Hemoperfusion allows mediator removal via absorption and is influenced by the interaction between the blood and the internal surface of the device, whose chemical characteristics may promote either selective and non-selective removal of mediators. 

### 7.1. Polymyxin B-Immoilized Cartridge (Toraymyxin)

This device consists of a column made by polystyrene fibers and Polymyxin-B, which is a cationic antibiotic binding endotoxin through ionic and hydrophobic forces. Each session lasts for 120 min and a second cycle has been repeated 24 h afterwards in most of the trials in this field [61]. Toraymyxin has been largely used in clinical practice [62,63], and some in vitro studies demonstrated significant Linezolid retention onto the internal surface of the device, while there was no significant removal of other antimicrobials (Meropenem, Imipenem/Cilastatin, Ceftazidime, Ceforozopram, Cefmetazole, Piperacillin, Ciprofloxacin and Vancomycin) [64,65]. In critical care practice, no significant influence on bloodstream concentration was demonstrated for Meropenem [66], although there is no evidence for other antimicrobials, thus warranting further investigation in this field. 

### 7.2. Porous Polystyrene Cartridge (Cytosorb)

This device contains a column made by highly porous polystyrene divinylbenzene copolymer covered with a biocompatible polyvinylpyrrolidone coating and is used for cytokine removal [12,44,60]. Hemoperfusion with Cytosorb may last 24 h and may be applied while RRT is running. A pre-clinical model on pigs [67] showed that Cytosorb significantly reduced the blood concentration of lipophilic antimicrobials in the range of 55 kDa (Fluconazole, Linezolid, Liposomal amphotericin, Posaconazole and Teicoplanin), while other agents were not influenced by this device (β-lactams: meropenem, piperacillin, ceftriaxone, flucloxacillin and cefepime; antifungals: anidulafungin; aminoglycosides: Tobramycin; quinolones: ciprofloxacin; macrolides: clarithromycin; lincosamides: clindamycin; azoles: metronidazole; antiviral: ganciclovir). Moreover, this study [67] observed that antimicrobial clearance was not stable over time, being highest at the beginning of this therapy. Although some studies have been performed to assess the safety and feasibility of Cytosorb in the management of critically ill septic patients [68,69], no research has ever systematically investigated the impact of this device on antimicrobial removal in daily practice, and the evidence available is restricted to case reports and small case series [70]. Recently, Scharf et al. [33] reported significant vancomycin adsorption with Cytosorb in seven critically ill septic patients, whose suboptimal bloodstream concentrations were attenuated by the two-hourly administration of 500 mg of this drug. In this setting, further clinical investigations are necessary to clarify the impact of Cytosorb on antimicrobial removal.

### 7.3. Microbind Affinity Blood Filter (Seraph 100)

The Seraph 100 Microbind Affinity Blood Filter (ExThera Medical, Martinez, CA, USA) is a sorbent made with polyethylene beads, whose internal surface has been modified to contain end-point attached heparin [71]. Hemoperfusion with Seraph 100 lasts 24 h and may be applied while RRT is running. The Seraph 100 has been recently introduced in critical care practice, and an in vitro study [72] investigated its absorptive properties on 18 antimicrobials (acyclovir, amphothericin B, ceftazidime, cefazolin, clindamycin, daptomycin, fluconazole, fosfomycin, gentamicin, levofloxacin, linezolid, meropenem, moxifloxacin, piperacillin, rifampicin, tazobactam, tobramycin and vancomycin) added to human donor plasma, showing that aminoglycosides were significantly removed during the first pass through the device. However, no clinical studies have addressed the issue of antimicrobial removal associated with the Seraph 100, and further investigation is warranted in this field.

## 8. Extracorporeal CO_2_ Removal and Extracorporeal Membrane Oxygenation

ECCO2-R is applied for the management of patients with hypercapnic respiratory failure (e.g., chronic obstructive pulmonary disease or acute respiratory distress syndrome [73,74]). ECMO is applied in patients with cardiopulmonary failure as: veno-venous ECMO, to provide extracorporeal CO_2_ removal and oxygenation in the context of Acute Respiratory Distress Syndrome (ARDS); veno-arterial ECMO, to provide organ perfusion in the context of acute cardiac dysfunction (e.g., septic cardiomyopathy) [75]. Although no data exist on antimicrobial removal during ECCO2-R, the delivery of ECMO is associated with Vd increase caused by the intravascular administration of priming solutions, fluid resuscitation and blood products that may alter the PK profile of hydrophilic antibiotics [34,35]. Furthermore, ECMO implies no pulsatile blood flow that may decrease the glomerular filtration rate and induce the renin–angiotensin–aldosterone axis activation, thus increasing fluid retention [34]. Moreover, ECMO may cause antimicrobial adsorption into the extracorporeal circuit, which is enhanced via a roller compared to centrifugal pumps [76], lipophilicity and high protein binding of the drugs [34,35]. 

Gomez et al. [35] reviewed antimicrobial removal in critically ill patients during ECMO. In vitro studies suggested large sequestration into the extracorporeal device of antimicrobials such as meropenem [77], cefazolin [76], teicoplanin [77], gentamicin [34], polymyxin B [77], vancomycin [78], micafungin [77], voriconazole [77], while this phenomenon was not reported for tigecycline [77], ciprofloxacin [79] and caspofungin [77]. In contrast, in vivo studies reported low bloodstream concentration during ECMO of teicoplanin [80] and voriconazole [81], while other antimicrobial concentrations were not significantly influenced by this device [20,82,83,84,85,86,87]. Recently, Shekar et al. investigated antimicrobial exposure in 85 patients who received ECMO, 38 (44.7%) of whom were on RRT. They found a large variation in the PK parameters and antimicrobial concentrations, with only 70 (56.5%) concentration profiles achieving the target exposure range. Poor target attainment was observed for oseltamivir (33.3%), piperacillin (44.4%) and vancomycin (27.3%). Accordingly, PK characteristics were reported highly variable in patients receiving ECMO, leading to low target exposure attainment when standard antimicrobial dosing regimens are used [88].

## 9. Effective Antimicrobial Dosing Strategies during ECOS

Appropriate dosing regimens are of paramount importance to achieve a therapeutic concentration of antimicrobials and allow microbiological eradication, contrast the emergence multidrug resistant pathogens [89] and prevent drug overdose and toxicity. In the ICU practice, a pharmacokinetic point-prevalence study [3] on standard antimicrobial dosing in 284 critically ill patients who received ß-lactams for infections, showed that 16% of them did not achieve 50% fT>MIC and that these patients were less likely to have a positive clinical outcome (odds ratio [OR], 0.68; *p* = 0.009). In contrast, positive clinical outcomes were associated with 50% fT>MIC and 100% fT>MIC ratios (OR, 1.02 and 1.56, respectively; *p* < 0.03) [3]. Moreover, this study reported large variability β-lactams plasma concentration, which was attenuated via a prolonged infusion (either extended infusion ≥2 h or continuous infusion) [3]. 

For this reason, several strategies for antimicrobial dosing have been implemented in clinical practice [16]:Unit-level interventions: this strategy includes the administration of antimicrobials driven by the PD characteristics of the drug, leading to the administration of time-dependent antimicrobials (β-lactams [90] and Linezolid [91]) via continuous infusion. However, no evidence exists on the implications of this strategy on the stability of antimicrobials bloodstream concentration during ECOS.Nomograms: this strategy accounts for patient-specific characteristics that may influence the PK profile of the drug (e.g., body weight and renal function [92,93]). Although single dosing nomograms have been implemented in clinical practice, they do not account for the concomitant delivery of ECOS.PK/PD-based antimicrobial dosing program: this strategy includes the use of software that performs a PK assessment based on patient-specific characteristics in conjunction with population pharmacokinetic models, via a Bayesian parametric approach and a Monte Carlo simulation [94]. However, this strategy has never been tested in critically ill patients who receive ECOS.Therapeutic Drug Monitoring (TDM): this strategy includes the evaluation of antimicrobial exposure via the assessment of drug concentration at the site of infection (e.g., blood, epithelial lining fluid or cerebrospinal fluid) and implies dose adjustment according to the pathogen susceptibility to the drug (MIC), in order to improve the PK/PD target attainment. In most of the cases, plasma concentration has been used as a surrogate for antimicrobial exposure at the source of infection [17]. TDM may play a role of paramount importance in the dose adjustment of antimicrobials with intra- and/or inter-individual PK variability and narrow therapeutic index [17], especially when PK characteristics are unknown or difficult to predict due to the patient’s clinical severity and instability. Accordingly, recently published guidelines recommended the TDM use for dose titration of ß-lactams, aminoglycosides, linezolid, teicoplanin, vancomycin and voriconazole [17]. In critically ill patients who receive RRT and ECMO [95,96,97], TDM has been largely used for adjusting the dose of ß-lactams, aminoglycosides, linezolid, teicoplanin, glycopeptides and colistin [17]. Furthermore, some evidence has been reported for daptomycin, fluoroquinolones and Tigecycline [17], although it warrants further investigation.

Currently, TDM has not been widely used in daily clinical practice and significant variation among hospitals was reported in terms of use, antimicrobial selection, sampling time points, assays, PK/PD targets and dose adjustment algorithms [98,99]. Moreover, critically ill septic patients receiving ECOS represent a subpopulation of patients that is commonly excluded from clinical investigations [51]. However, this tool may enable real-time antimicrobial PK measurements at the bedside, thus allowing clinicians to adjust the antimicrobial dose to the evolving patient’s clinical condition and pathogen susceptibility. The use of chromatography-based TDM should be fostered in daily large-scale clinical practice to provide a rapid and reliable detection of antimicrobial concentrations in plasma and tissues (e.g., alveola lining fluid), in order to guide antimicrobial dose titration aiming to improve clinical and microbiological outcomes, even in remote and resource-limited settings [17]. 

## 10. Conclusions

In critically ill septic patients who receive ECOS, antimicrobial exposure is highly-variable and may lead to poor target attainment with consequently worse clinical outcomes. In comparison with “standard” dosing regimens, several strategies have been proposed to improve antimicrobial exposure, although low evidence supports their use in daily clinical practice. Among them, the therapeutic drug monitoring appears as a promising tool to efficiently measure the antimicrobial PK and adapt antimicrobial dosing to pathogen susceptibility, in order to target an adequate exposure via a precision medicine and personalized approach. For this reason, further large-scale investigation is necessary to place a role for therapeutic drug monitoring in the management of critically ill septic patients receiving ECOS in daily clinical practice.

## Figures and Tables

**Figure 1 microorganisms-11-00473-f001:**
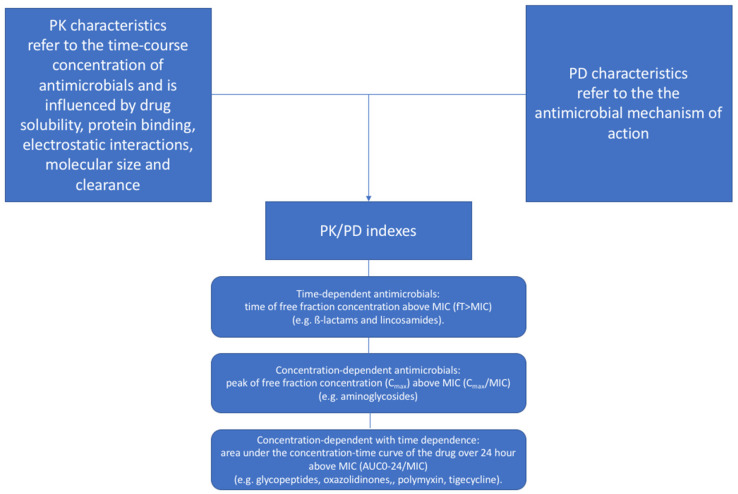
PK/PD indexes. Abbreviations: AUC, area under curve; MIC, minimum inhibitory concentration; PD, pharmacodynamic; PK, pharmacokinetic.

**Table 1 microorganisms-11-00473-t001:** PK/PD characteristics of antimicrobials.

Antimicrobials	PK/PD Index	Free Fraction%	Volume of Distribution (L kg^−1^)	Route of Elimination
**ANTIBIOTICS**				
β-***lactams***				
Amoxicillin/clavulanate	fT>MIC	82/75	0.36/0.21	R/L
Piperacillin/Tazobactam	fT>MIC	70/78	0.24/0.40	R
Oxacillin	fT>MIC	6–10	0.4	R
Ceftriaxone	fT>MIC	10	0.1–0.2	R/L
Cefepime	fT>MIC	84	0.3	R
Ceftaroline	fT>MIC	80	0.29	R
Ceftazidime	fT>MIC	90	0.28–0.40	R
Ceftazidime/Avibactam	fT>MIC	90/92	0.28/0.31	R
Ceftolozane/Tazobactam	fT>MIC	80/78	0.19/0.40	R
Cefiderocol	fT>MIC	60	0.26	R
Meropenem	fT>MIC	98	0.35	R
Imipenem/Cilastatin	fT>MIC	80/56	0.22–0.24	R
Meropenem/Vaborbactam	fT>MIC	98/77	0.28/0.25	R
** *Aminoglycosides* **				
Amikacin	C_max_/MIC	>95	0.22–0.5	R
Gentamicin	C_max_/MIC	>95	0.36	R
** *Glyco-, glycolipo- and Lipopeptides* **				
Daptomycin	AUC_24_/MIC	20	0.1–0.13	R
Teicoplanin	AUC_24_/MIC	10–40	0.5–1.2	R
Vancomycin	AUC_24_/MIC	50–90	0.47–1.1	R
** *Glycylcycline* **				
Tigecycline	AUC_24_/MIC	11–29	0.12	L
** *Lincosamides* **				
Clindamycin	AUC_24_/MIC	5–15	1.1	L
** *Macrolides* **				
Azithromycin	AUC_24_/MIC	50–93	0.47	L
** *Monobactam* **				
Aztreonam	fT>MIC	44	0.18	R
** *Nitroimidazoles* **				
Metronidazole	AUC_24_/MIC	80	0.6–0.85	R
** *Oxazolidinones* **				
Linezolid	AUC_24_/MIC	70	0.5–0.8	L
Tedizolid	AUC_24_/MIC	50–90	0.95–1.14	L
** *Polymyxins* **				
Colistin	AUC_24_/MIC	59–74	0.3–0.4	R
** *Quinolones* **				
Ciprofloxacin	AUC_24_/MIC	60–80	2.5	R/L
Levofloxacin	AUC_24_/MIC	60–75	1.1–1.5	R
** *Rifamycins* **				
Rifampin	AUC_24_/MIC	20	0.65	R/L
** *Tetracyclines* **				
Doxycycline	AUC_24_/MIC	7	0.75–1.91	R/L
**ANTIMYCOTICS**				
Liposomal Amphotericin B	C_max_/MIC	10	4	R/L
Fluconazole	AUC_24_/MIC	88	0.7	R
Isavuconazole	AUC_24_/MIC	<1	6.42	R/L
Itraconazole	AUC_24_/MIC	<1	0.14	L
Posaconazole	AUC_24_/MIC	<1	3.22–4.21	L
Voriconazole	AUC_24_/MIC	40	4.6	L
Anidulafungin	AUC_24_/MIC	<1	0.4–0.7	L
Caspofungin	AUC_24_/MIC	3	0.11	L
**ANTIVIRALS**				
Acyclovir	-	91–67	0.7	R
Ganciclovir	-	99–98	0.7	R
Oseltamivir	-	97	23–26	R
Remdesivir	-	<20	2.05	R

Abbreviations: L, liver; MIC, minimum inhibitory concentration; R, renal; fT>MIC, time of free fraction concentration above MIC (time-dependent antimicrobials); C_max_/MIC, peak of maximum concentration above MIC (concentration dependent antimicrobials); AUC_24_/MIC, area under the concentration–time curve over 24 h above MIC (concentration-dependent with time dependence antimicrobials).

**Table 2 microorganisms-11-00473-t002:** Extracorporeal Organ Support Therapy.

Extracorporeal Organ Support Therapy				References
**Extracorporeal Blood Purification Therapies**	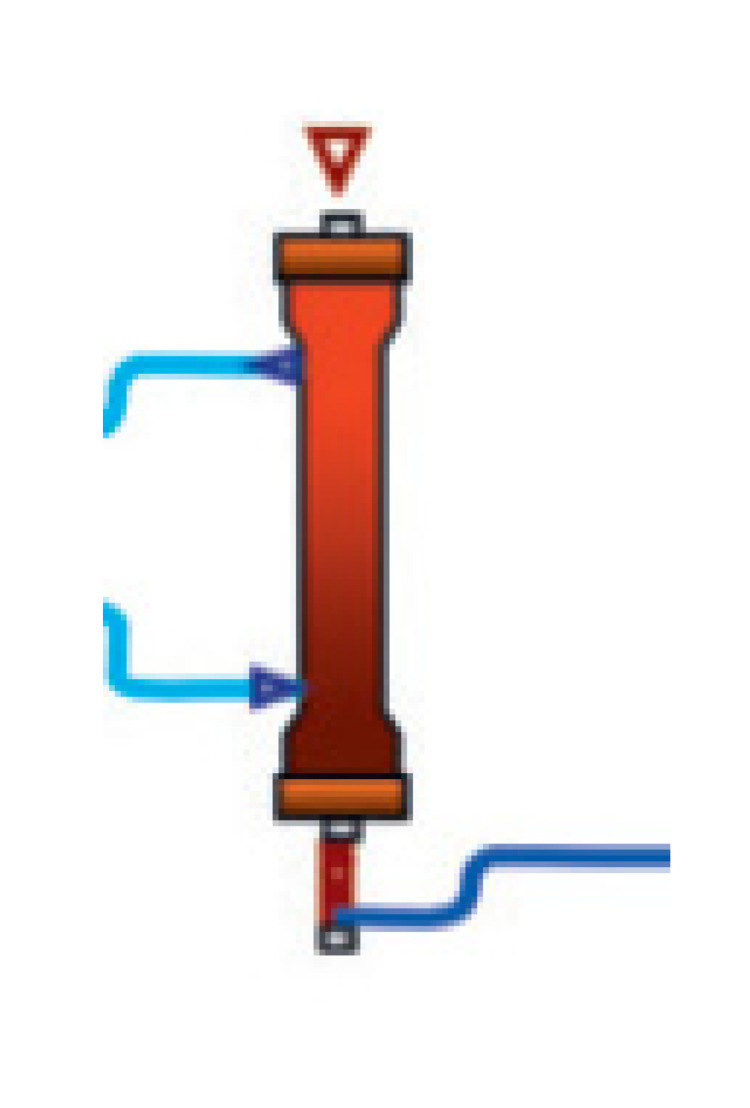	**CRRT**	Continuous Renal Replacement Therapy (CRRT) allows blood purification from the life-threatening waste product overload occurring during acute kidney injury. The three main mechanisms of solute and fluid removal during CRRT are: − continuous venous–venous haemodialysis (CVVHD), trans-membrane solute removal via concentration gradient (diffusion). − continuous venous–venous haemofiltration (CVVHF), trans-membrane water and solute removal via pressure gradient (filtration). − continuous venous–venous haemodiafiltration (CVVHDF), combines CVVHD and CVVHF. Antimicrobial PK characteristics that may favour drug removal during CRRT are: hydrophilicity, small molecular weight (<500–1000 Da), low protein binding (<80%) and small Vd (<2 L/Kg).	[29,30]
	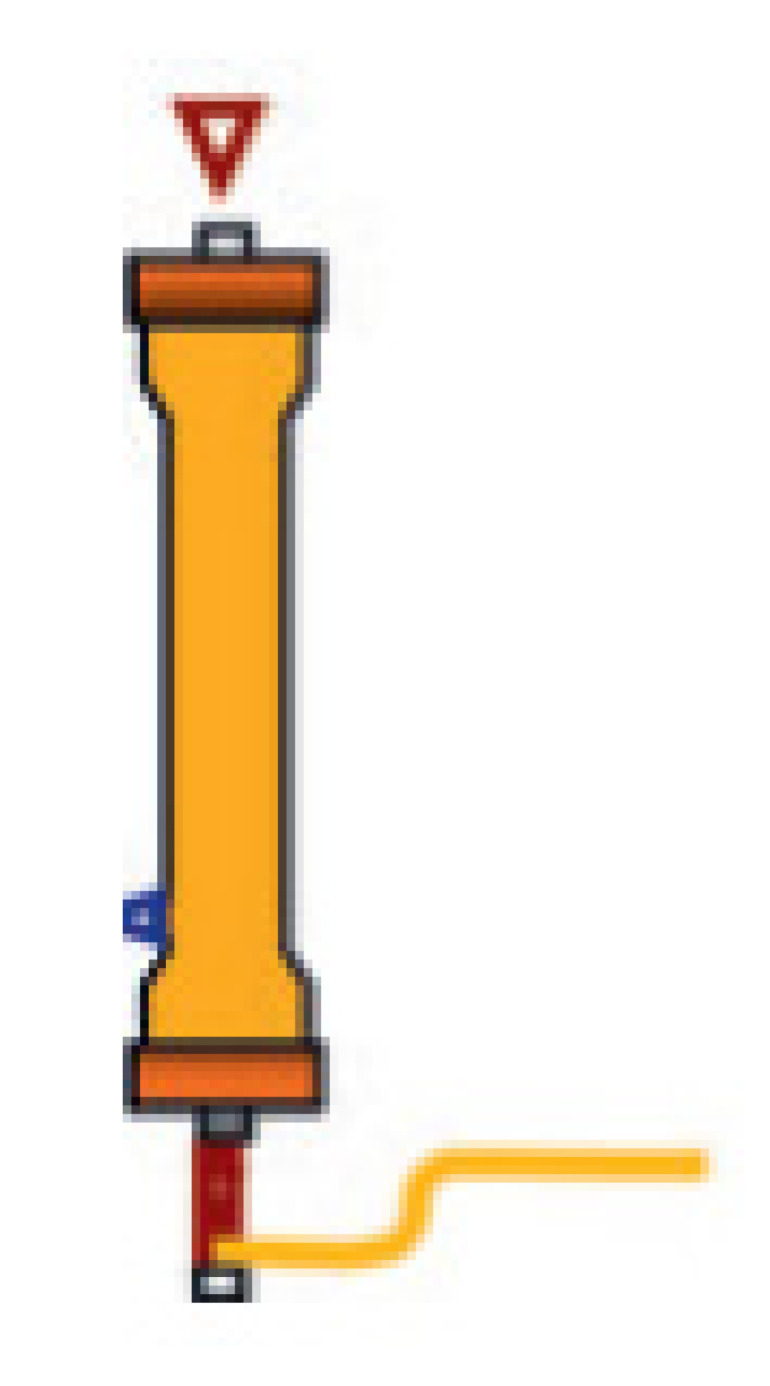	TPE	Therapeutic plasma exchange (TPE) allows plasma filtration via high cut-off membranes and subsequent replacement with solutions of donor plasma, colloids, crystalloids or a mixture thereof. Main indications for this therapy are represented by immunopathological conditions such as myasthenia gravis, Guillain–Barré syndrome and Waldenström macroglobulinemia. However, TPE has been used in patients with sepsis, although no definitive evidence exists in this field.Antimicrobial PK characteristics that may favour drug removal during TPE are low Vd and high protein binding (>80%).	[31]
	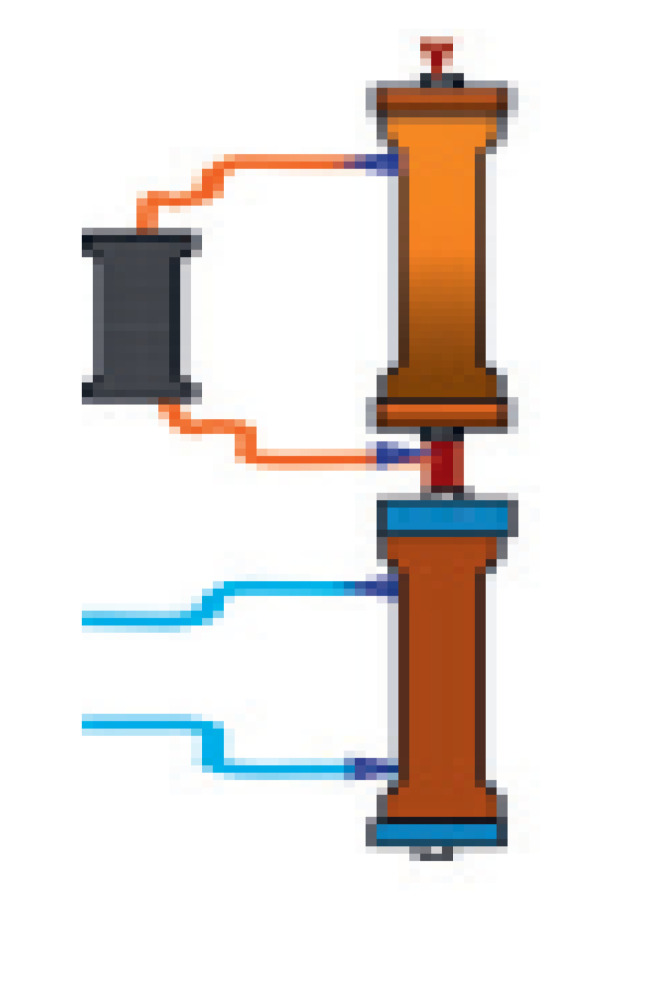	CPFA	Coupled plasma filtration and adsorption (CPFA) combines plasma filtration via high cut-off membranes with subsequent adsorption via styrene resin. The volume plasma purified by waste products is then reinfused into the blood line and the whole blood passes across an hemofilter for further solute removal. This therapy has been used in septic patents, although clear benefit has never been demonstrated.CPFA was demonstrated to significantly lower the bloodstream concentration of colistin, whose amount was directly proportional to the volume of plasma filtered over time.	[32]
	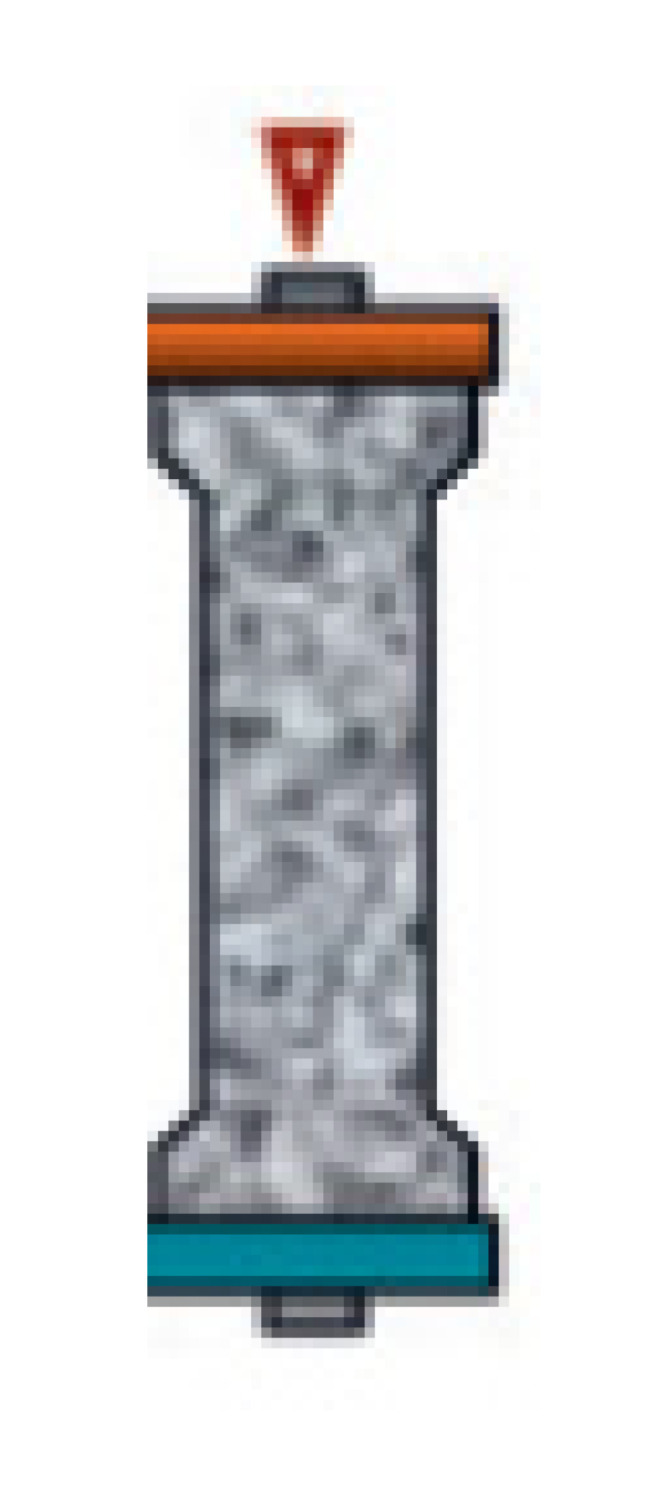	HP	Hemoperfusion allows extracorporeal removal of mediators via absorption and, according to cartridge characteristics, may be classified as: *Selective* ⇒ Endotoxin removal via Polymyxin B-immobilized cartridge (Toraymyxin); no evidence of significant antimicrobial removal in vivo. *Non selective* ⇒ Cytokine removal via porous polystyrene cartridge (Cytosorb); small case series reported significant absorption of vancomycin. ⇒ Pathogen removal via microbind affinity blood filter (Seraph 100); no evidence of significant antimicrobial removal in vivo.	[33]
**Cardio-pulmonary support**	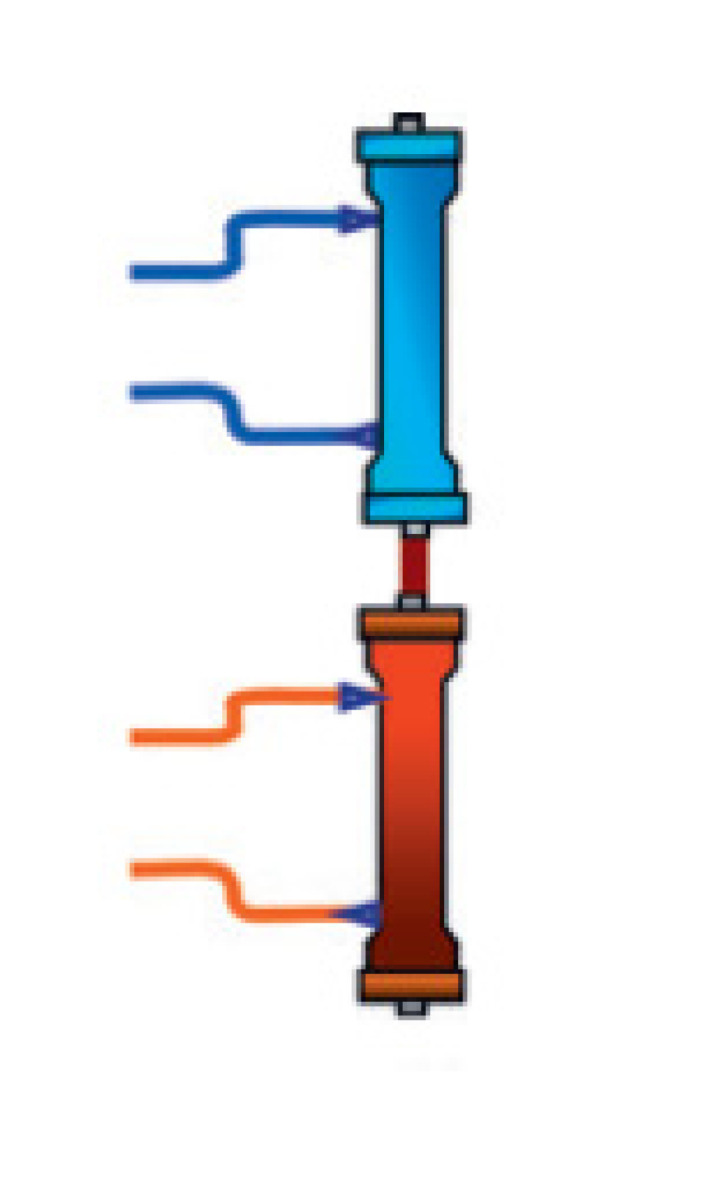	ECCO2R	Extracorporeal CO_2_ removal (ECCO2R) allows CO_2_ removal in hypercapnic respiratory diseases (e.g., COPD and ARDS); no evidence of significant antimicrobial removal in vivo.	
	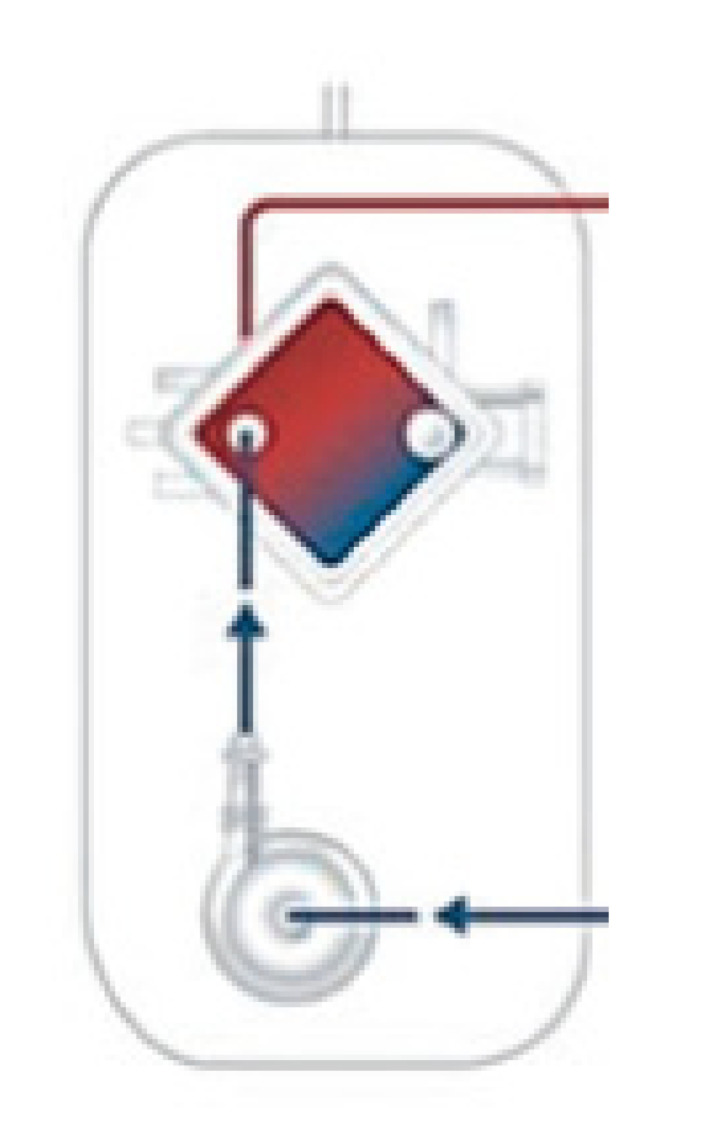	ECMO	Extracorporeal membrane oxygenation (ECMO) may be set as: ⇒ Veno-venous ECMO, that allows blood oxygenation and CO_2_ removal, and is indicated in patients affected by acute severe gas exchange alterations (e.g., ARDS); ⇒ Veno-arterial ECMO, that allows cardiac output replacement and is indicated in patients with cardiogenic shock. Antimicrobial PK characteristics that may favour drug removal during ECMO are lipophilicity and high protein binding (>80%).	[34,35]

Abbreviations: ARDS, acute respiratory distress syndrome; COPD, chronic obstructive pulmonary disease; CPFA, continuous plasma filtration and absorption; CRRT, continuous renal replacement therapy; ECCO2R, extracorporeal carbon dioxide removal; ECMO, extracorporeal membrane oxygenation; HP, hemoperfusion; PK, pharmacokinetic; TPE, therapeutic plasma exchange; Vd, volume of distribution.

## Data Availability

Not applicable.

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
