# Peer review of "Antimicrobial Exposure in Critically Ill Patients with Sepsis-Associated Multi-Organ Dysfunction Requiring Extracorporeal Organ Support: A Narrative Review"

_microorganisms, 2023, doi:10.3390/microorganisms11020473_

Round 1

Reviewer 1 Report

The manuscript entitled "Antimicrobial exposure in critically ill patients with sepsis-associated multi-organ dysfunction requiring extracorporeal organ support: a narrative review" highlights the appropriate use of antimicrobials in relation to extracorporeal organ support to minimize infections leading to sepsis. 

The article is written well and also shows the significance of contents in the relevant field. However, it is suggested to improve the conclusion section to support the important findings of the manuscript. Furthermore, major English editing is required to improve the manuscript. 

Author Response

Dear reviewer,

We thank you very much for your important feedback. Please, see below our point-by-point response:

The manuscript entitled "Antimicrobial exposure in critically ill patients with sepsis-associated multi-organ dysfunction requiring extracorporeal organ support: a narrative review" highlights the appropriate use of antimicrobials in relation to extracorporeal organ support to minimize infections leading to sepsis. 

The article is written well and also shows the significance of contents in the relevant field. However, it is suggested to improve the conclusion section to support the important findings of the manuscript. Furthermore, major English editing is required to improve the manuscript. 

  • Thanks for your comments. We improved the conclusion section as you recommended. Moreover, we performed an extensive English revision as kindly you advised.

Reviewer 2 Report

This is an interesting review in regard to a very critical issue in clinical practice. The authors present the importance of this issue, which is the antimicrobial exposure in patients receiving extracorporeal organ support. It's indeed important to warrant further investigation, thereby adjusting the doses of antibiotic and remedial model. Any breakthrough in this field is expected. 

There are several key viewpoints that the authors want to emphasize. For example, there is consideration of side effects during antimicrobial exposure for the sepsis patients; sepsis patients may have special PK/PD profiles. Extracorporeal organ support changes the PK/ PD profiles and increase the clearance of drugs, in most instances.

Unfortunately, these viewpoints that continue to distress clinicians have existed for a long history. Plenty of related reviews have been published. However, there is no remarkable recent advance elucidated in this review. Furthermore, the authors have not provided novel viewpoints to solve these clinical trouble. Therefore, this review may be less attractive or innovative unless more novel or detailed information is reviewed. For instance, when the authors focused on CRRT therapy, they may provide different kinds of antibiotics PK/PD characteristics in this condition, even recommendation that favors clinicians to adjust the doses. Furthermore, the authors provide dosing strategies as the last part of this review, which may be the key solution to this clinical problem. However, this part of context is relatively brief, suggesting no promise to clinicians. Therefore, these details may be considered to further improve this review.

Author Response

Dear Reviewer,

We thank you very much for your important feedback. Please, see below our point-by-point response:

This is an interesting review in regard to a very critical issue in clinical practice. The authors present the importance of this issue, which is the antimicrobial exposure in patients receiving extracorporeal organ support. It's indeed important to warrant further investigation, thereby adjusting the doses of antibiotic and remedial model. Any breakthrough in this field is expected. 

There are several key viewpoints that the authors want to emphasize. For example, there is consideration of side effects during antimicrobial exposure for the sepsis patients; sepsis patients may have special PK/PD profiles. Extracorporeal organ support changes the PK/ PD profiles and increase the clearance of drugs, in most instances.

Unfortunately, these viewpoints that continue to distress clinicians have existed for a long history. Plenty of related reviews have been published. However, there is no remarkable recent advance elucidated in this review. Furthermore, the authors have not provided novel viewpoints to solve these clinical trouble. Therefore, this review may be less attractive or innovative unless more novel or detailed information is reviewed. For instance, when the authors focused on CRRT therapy, they may provide different kinds of antibiotics PK/PD characteristics in this condition, even recommendation that favors clinicians to adjust the doses. Furthermore, the authors provide dosing strategies as the last part of this review, which may be the key solution to this clinical problem. However, this part of context is relatively brief, suggesting no promise to clinicians. Therefore, these details may be considered to further improve this review.

  • Thanks for your comments. We focused on specific PK/PD antimicrobial characteristics during CRRT and added some recommendations for antimicrobial dose titration in this setting. Moreover, we improved the section on dosing strategies and highlighted the important role covered by therapeutic drug monitoring to improve microbiological and clinical outcomes. Moreover, we performed an extensive English revision as kindly you advised.

Round 2

Reviewer 2 Report

The revised version along with a cover letter has been received. The authors improved the section on dosing strategies and therapeutic drug monitoring. Moreover, they kindly improved the extensive English revision. The quality of the manuscript has remarkably been raised. 

Unfortunately, the key concerns I had were not well addressed. For example, the authors strengthened the therapeutic drug monitoring and enriched their description. However, the revised content still failed to answer the question raised by many clinicans, as no novel contribution to this field was made. There is no doubt that this unpleasant regret is because of the lack of new advance in this field. But it still make this review appear a little less innovative, at least not as good as many other related articles. Personally, I feel that the quality of the manuscript is a bit difficult to radically improve based on the existing framework. Therefore, I humildly advise editors to consider about rejections of this manuscript.

Author Response

The revised version along with a cover letter has been received. The authors improved the section on dosing strategies and therapeutic drug monitoring. Moreover, they kindly improved the extensive English revision. The quality of the manuscript has remarkably been raised. 

Unfortunately, the key concerns I had were not well addressed. For example, the authors strengthened the therapeutic drug monitoring and enriched their description. However, the revised content still failed to answer the question raised by many clinicans, as no novel contribution to this field was made. There is no doubt that this unpleasant regret is because of the lack of new advance in this field. But it still make this review appear a little less innovative, at least not as good as many other related articles. Personally, I feel that the quality of the manuscript is a bit difficult to radically improve based on the existing framework. Therefore, I humildly advise editors to consider about rejections of this manuscript.

R: Thanks for your comment. The use of TDM in patients who receive ECOS is evolving and we lack good quality evidence. Although it was not the main topic of our manuscript, we improved further this section. In comparison with other papers, we provided an updated and comprehensive review of mechanisms of antimicrobial removal in the heterogeneous world of ECOS. Moreover, we discussed potential application of non-standard dosing strategies that may help the physician to improve patient care.